# A unified metric of generalization across humans and machines

## Abstract

Generalization means performing well on situations that differ from those seen during learning. Accuracy alone cannot tell whether a system truly generalizes, because a model can be correct yet fragile or misaligned with the structure of a task Ilievski et al. (2025). We introduce $GR^\star$, a single reproducible metric that measures not only performance but also stability and structural alignment while accounting for data, scale, and abstraction cost. $GR^\star$ is designed to be simple, deterministic, and fair across humans and machines, allowing both to be compared under the same coordinate system. All evaluations follow a lightweight standardized pipeline with fixed hyperparameters and no distributed training, ensuring transparency and reproducibility. This work turns generalization from an abstract concept into a measurable and falsifiable property, offering a unified and interpretable way to understand how different systems learn. Code: `https://anonymous.4open.science/r/GR_Star_Unified_Metric-8E84/README.md`. Dataset: `https://scikit-learn.org/stable/datasets/toy_dataset.html`

## 1 Introduction

Artificial intelligence has advanced from domain heuristics to general learning systems, that accurately model biomolecular interactions Abramson et al. (2024), achieve and surpass world-champion play in Go Silver et al. (2016) and StarCraft II Vinyals et al. (2019), win head-to-head real-world drone races Kaufmann et al. (2023), coordinate at human level in multiagent 3D environments Jaderberg et al. (2019), master imperfect-information strategy Perolat et al. (2022), master chess, shogi and Go through self-play with a single algorithm Silver et al. (2018), generalize across hundreds of control tasks via world models Hafner et al. (2025), support structured gameplay ideation with world and human action models Kanervisto et al. (2025), and elicit verifiable reasoning in large language models (LLMs) Guo et al. (2025), establishing a scalable trajectory across perception, control and cognition. Despite these advances, accuracy alone cannot tell whether a system truly learns or merely fits a benchmark. A reliable measure of generalization must reveal how stable a learner remains when conditions shift, how well its internal structure aligns with the task, and how efficiently it achieves that performance. We aim to turn generalization from an abstract idea into something measurable, interpretable, and reproducible. However, when a task changes even slightly—such as switching from clean to noisy data Hendrycks & Dietterich (2019), altering class balance Liu et al. (2021), or updating prompts Gonen et al. (2023) and scoring rules-systems that appear strong on one benchmark often lose accuracy or even reverse rank with simpler baselines, because plain accuracy hides instability and structural mismatch and fails to transfer between human reasoning and machine learning settings. This brittleness poses a bottleneck for practical use and fair comparison, as the field lacks a model-agnostic, falsifiable, and portable generalization measure that can separate accuracy from stability and structural alignment while normalizing sample size, feature scale, and abstraction cost into a single comparable scale across humans and machines Mitchell et al. (2019); Pushkarna et al. (2022); Kapoor et al. (2024). Creating a unified, model-agnostic metric of generalization shared by humans and machines would provide a single reproducible scale for evaluation, enable fair comparison across capacities and domains, make brittleness falsifiable before deployment, and guide data and model choices toward systems that remain stable and transferable across tasks.

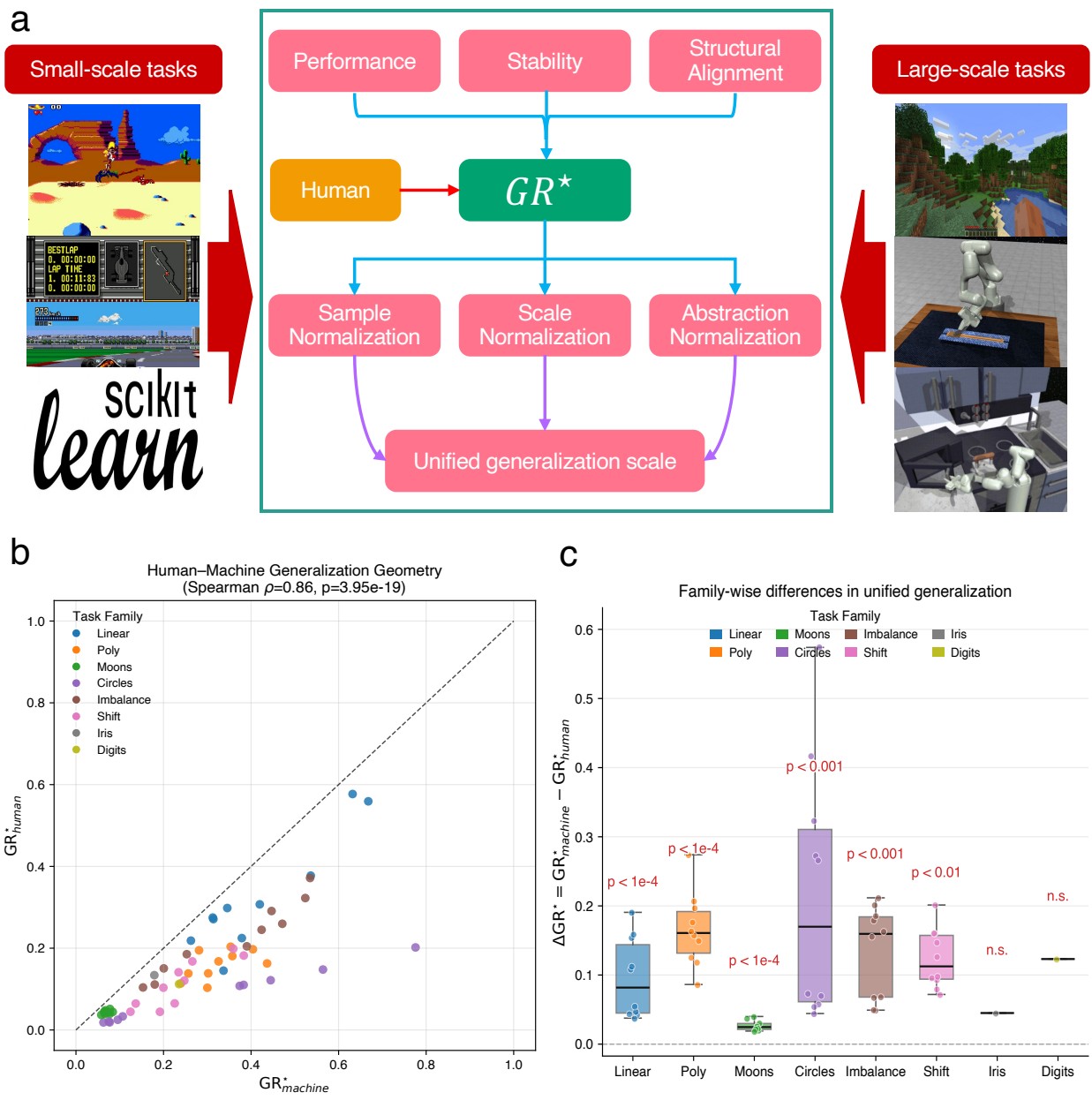

Figure 1: Unified generalization geometry across humans and machines. (a) Computation of $GR^\star$ combines measured performance, stability and structural alignment with normalization by sample size, feature scale and abstraction cost. (b) Human and machine scores for each task are plotted on the same axis, colors denote families and the diagonal gives a reference for equality. (c) Family level distributions of $\Delta GR^\star$ are shown as boxplots with jittered points and $p$ values adjusted for the false discovery rate.

Here, we present a unified, model-agnostic metric of generalization that places human heuristics and machine learners on a common scale by normalizing performance, stability, and structural alignment with respect to sample size, feature scale, and abstraction cost. This enables portable, falsifiable, and low-compute evaluation for fair comparison and reproducible practice across tasks. The metric is based on a common coordinate system that maps humans and machines by decomposing generalization into performance, stability, and structural alignment; each component is normalized by sample size, feature scale, and abstraction cost, yielding a portable, falsifiable, model-agnostic score across tasks and compute and data budgets. This

provides the capability for fair comparison and reproducible evaluation Lu et al. (2019); Zhang & Yang (2021); Verwimp & et al. (2024). As shown in Figure. 1, the metric multiplies task performance by a stability factor, includes a structural alignment term, and normalizes the result by sample size, feature scale, and abstraction cost to yield a single comparable score, and the same standardized procedure is applied to human heuristics and learning models. Although progress has been rapid, a unified model-agnostic metric that jointly captures performance, stability, and structural alignment, normalizes sample size, feature scale, and abstraction cost, and enables fair comparison between humans and machines is still lacking Tenenbaum et al. (2011). We solve this with a model-agnostic metric and a standardized pipeline that normalize performance, stability, and structural alignment by sample size, feature scale, and abstraction cost for fair and reproducible comparison across humans and machines. Validation spans 62 tasks across synthetic and real datasets, using fixed hyperparameters and seeds under a standardized protocol on a single Apple M4 Max, and the same evaluation applied to human heuristics and learning models reveals a consistent geometry of generalization with Spearman $\rho = 0.86$ and exact reproducibility. On the unified scale, tasks cluster by family, with explicitly structured tasks scoring higher, nonlinear or shifted regimes scoring lower and showing greater variability, and the metric detecting robustness degradation under spurious correlations before accuracy declines.

To compare humans and machines fairly, we propose a model agnostic metric of generalization that remains robust under shifts in data or tasks and unifies performance, stability, and structural alignment Bousquet & Elisseeff (2000). Measuring generalization across humans and machines is difficult because accuracy, stability, and structural alignment are entangled, and small shifts in data or scoring flip rankings, as seen in recent analyses of distributional shifts and benchmark drift Li & Flanigan (2024); Lewis & Mitchell (2024). Existing approaches depend on task specific benchmarks and ad hoc metrics, conflate accuracy with stability and structural alignment, and suffer data contamination and benchmark drift, which prevents portable and reproducible comparison between humans and machines Mundt et al. (2023); Koh et al. (2021); Recht et al. (2019); Deng et al. (2024). Applied out of the box, GR$^\star$ unifies performance, stability, and structural alignment into a single generalization score comparable across humans and machines.

In this work, we introduce GR$^\star$, a unified, model-agnostic generalization metric that enables portable, falsifiable, and reproducible comparison between humans and machines across tasks, data regimes, and compute budgets. Specifically, this study makes three primary contributions:

1. proposes a unified generalization score that integrates performance, stability, and structural alignment, normalized by sample size, feature scale, and abstraction cost, providing a single interpretable scale shared by humans and machines;

2. establishes a standardized low-compute evaluation pipeline with fixed hyperparameters and seeds, covering 62 tasks across synthetic and real datasets on a single Apple M4 Max, yielding exact reproducibility and a consistent geometry of generalization with Spearman $\rho = 0.86$;

3. demonstrates early brittleness detection and falsifiability, where GR$^\star$ detects robustness degradation under spurious correlations before accuracy declines, remains directionally stable within task families, and is sensitive to true structural signal while robust to non-structural perturbations.

## 2 A Unified Human and Machine Generalization Geometry

We aim to place humanlike, low-capacity learners Holzinger & et al. (2023); Gulwani & et al. (2015) and standard machine learners on a single generalization scale so that cross-system structure becomes visible and testable. Accuracy alone confounds instability and resource effects, therefore we design a metric that rewards reliable performance, reflects representational alignment, and discounts the cost of achieving it. This approach is conceptually related to the generalized out-of-distribution framework Yang et al. (2024), which unifies diverse evaluation tasks under one view, but here our goal is to quantify structural and resource-normalized generalization rather than detection. The design and intended data flow are summarized in the schematic panel (Figure. 1a; Table. 1). We define the metric GR$^\star$ by combining performance, stability, and structural alignment while normalizing by sample, model scale, and abstraction costs as:

Table 1: Formal definition of the unified generalization metric GR$^\star$. Each term in the formula reflects a specific source of divergence between human and machine generalization identified in prior literature. The sample normalization and scale normalization terms relate to factors partially addressed in classical generalization metrics (e.g., sample complexity, capacity, and norm-based bounds), whereas the alignment term and abstraction cost are introduced in this work as they capture structural alignment and abstraction-driven generalization—two dimensions highlighted in human–machine generalization research but absent from existing metrics. By integrating all four dimensions into a single coordinate system, GR$^\star$ offers a unified and interpretable way to quantify human–machine generalization differences.

| Symbol | Definition | Intuition |
|---|---|---|
| Acc | Empirical task accuracy. | The primary signal of success on the given task. |
| $U = 2\sqrt{\text{Acc}(1-\text{Acc})}$ | Derived uncertainty term. | Peaks when accuracy $= 0.5$; the stability factor $(1-U)$ downweights unstable mid-range accuracy. |
| $\tau + D_{\text{align}}$ | Alignment term between learner's internal relevance vector and a monotonic template mapped to $[0,1]$. | Rewards learners whose internal representations follow the structure of the task instead of memorizing labels. |
| $c_0$ | Baseline constant $= 0.15$. | Prevents division by very small denominators and anchors the metric scale. |
| $\tilde{N} = n \div 1000$ | Normalized sample count. | More data increases cost; normalization discourages over-reliance on data quantity. |
| $\tilde{S} = s \div 100$ | Normalized feature or parameter scale. | Captures model capacity; penalizes performance gained purely by scaling up. |
| $\beta\sqrt{\tilde{S}}$ | Sublinear scale normalization. | Doubling model size increases cost but slower than linearly. |
| $\gamma\bar{D}$ | Abstraction cost derived from the entropy of the relevance structure. | Higher when the learner depends on abstract or diffuse features, lower when it captures the task structure directly. |
| $\alpha, \beta, \gamma$ | Global fixed weights (1.0, 0.5, 1.0). | Keep cost trade-offs consistent and deterministic across tasks. |
| $\tau = 0.20$ | Minimum alignment mass. | Guarantees non-zero structure alignment even for simple or nearly separable tasks. |

$$\text{GR}^\star = \frac{\overbrace{\text{Acc}}^{\text{Performance}} \cdot \underbrace{(1-U)}_{\text{Stability}} \cdot \underbrace{(\tau + D_{\text{align}})}_{\text{Structural Alignment}}}{\underbrace{c_0}_{\text{Baseline Cost}} + \underbrace{\alpha \log(1+\tilde{N})}_{\text{Sample Cost}} + \underbrace{\beta\sqrt{\tilde{S}}}_{\text{Scale Cost}} + \underbrace{\gamma\tilde{D}}_{\text{Abstraction Cost}}},$$

$$U = 2\sqrt{\text{Acc}(1-\text{Acc})}.$$

(1)

Hyperparameters are fixed across all tasks as $\alpha = 1.0$, $\beta = 0.5$, $\gamma = 1.0$, $\tau = 0.20$, $c_0 = 0.15$. Deterministic normalizations are used throughout, with $\tilde{N} = n/1000$ for sample count $n$, $\tilde{S} = s/100$ for feature dimension $s$, and $\tilde{D}$ defined as the entropy of a one-dimensional feature–label relevance vector normalized by its maximum; the alignment term $D_{\text{align}}$ is a centered cosine between the normalized relevance vector and a fixed monotone template mapped to $[0,1]$. No task-specific tuning is applied. To minimize implementation variance and foreground reproducibility, all tasks are instantiated through a rigorously deterministic open-source reference

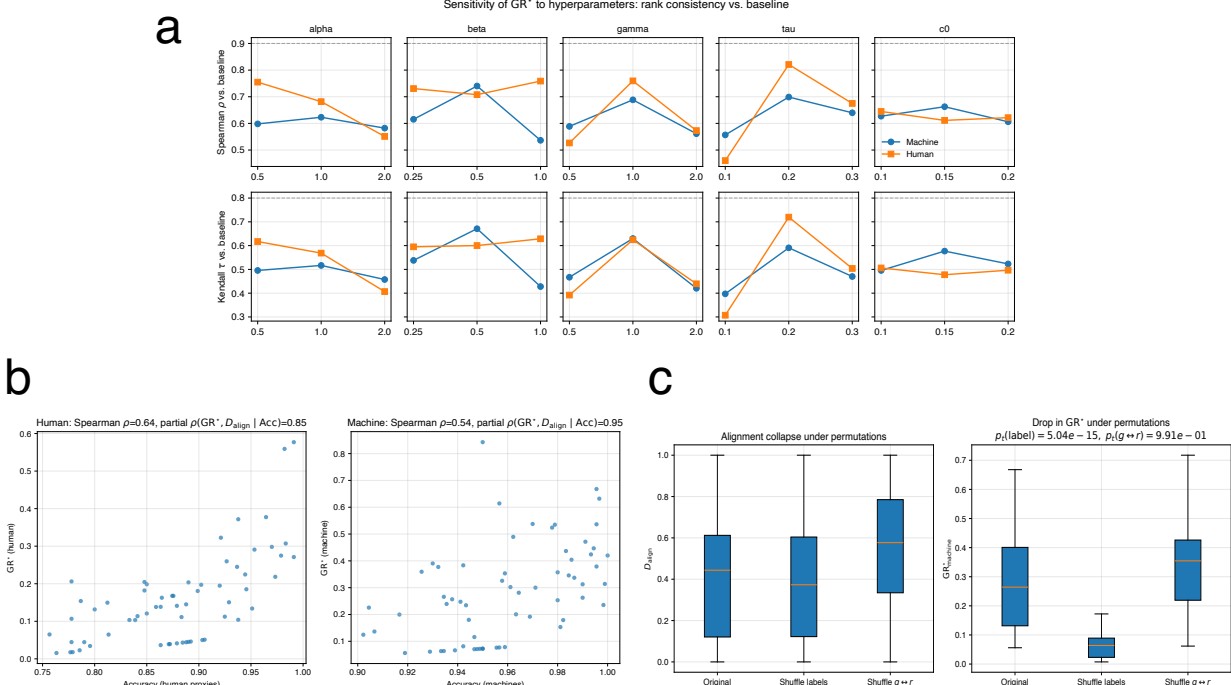

Figure 2: Robustness, independence, and falsification of GR$^\star$. (a) Rank stability under parameter sweeps. (b) Information beyond accuracy via partial correlations. (c) Collapse only when true label–feature structure is destroyed.

pipeline that leverages standardized scikit-learn interfaces as a stable substrate; seeds are fixed, execution is scripted in a single order, and inputs are standardized within each task before training. Human proxies are three low-capacity, interpretable learners, $L_1$-regularized logistic regression, three-neighbor k-nearest neighbors, and Gaussian naive Bayes. Machine learners are support vector classification with an RBF kernel, random forests with maximum depth eight using task-indexed seeds, and a multilayer perceptron with two hidden layers of 64 units each using the same seeding policy. We evaluate sixty-two tasks across eight families—Linear, Poly, Moons, Circles, Imbalance, Shift, and the real datasets Iris and Digits—and for every task we compute GR$^\star_{\mathrm{human}}$ and GR$^\star_{\mathrm{machine}}$. Across the sixty-two tasks the Spearman rank correlation between systems is $\rho = 0.86$ with $p = 3.95 \times 10^{-19}$. Points concentrate near the diagonal with clear family stratification; Linear and Imbalance tend to lie higher on the unified scale, Moons, Circles, and Shift occupy lower regions, and Iris and Digits fall on the same continuum rather than as outliers. The mild leftward concentration reflects cost normalization because the denominator penalizes sample scarcity, model scale, and abstraction. The pipeline is fully deterministic and reruns yield identical statistics. These observations establish a stable geometric coordinate shared by human proxies and machine learners for ordering task difficulty, which is the central finding of this panel (Figure. 1b).

## 3 Family Level Differences and Structural Alignment

Having established that humans and machines share a consistent geometry on the unified scale, we next quantify directional differences. For each task, the difference between the machine and human scores is defined as:

$$\Delta \mathrm{GR}^\star = \mathrm{GR}^\star_{\mathrm{machine}} - \mathrm{GR}^\star_{\mathrm{human}}. \tag{2}$$

Positive values indicate a machine advantage on the same coordinate system. The evaluation spans eight families under a single deterministic script with fixed seeds and a fixed order. Ten tasks are generated for each

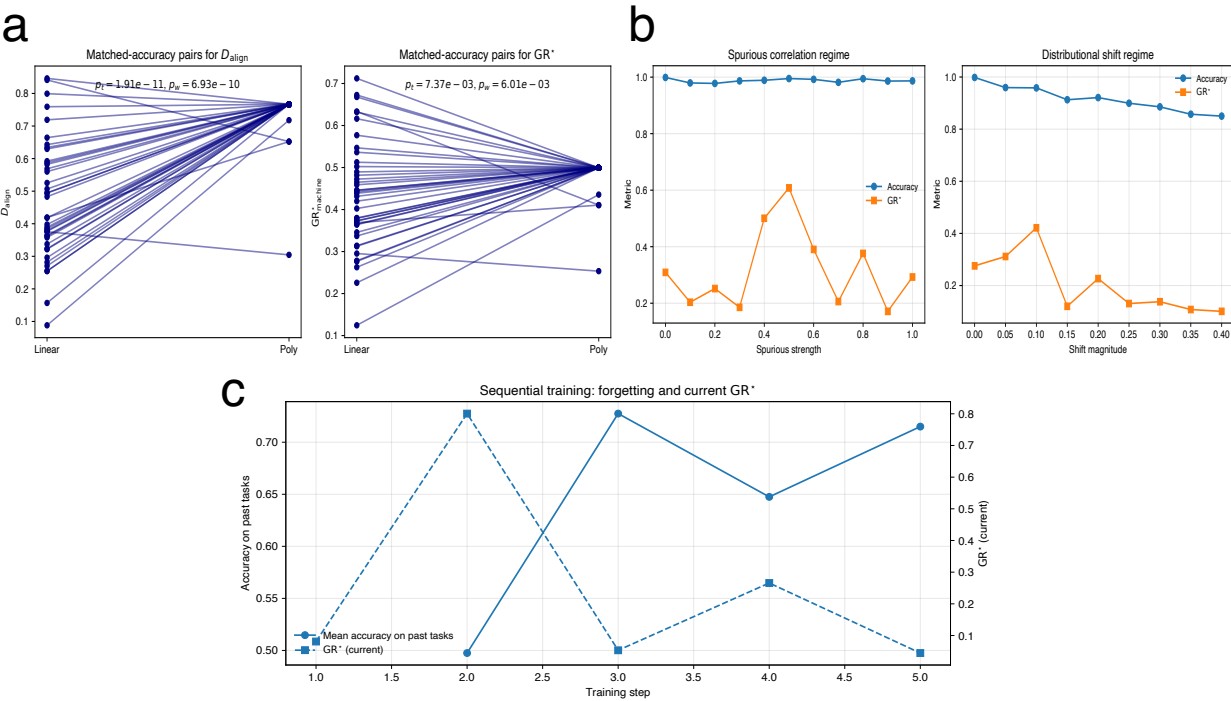

Figure 3: Structural independence, early brittleness detection, and sequential dynamics. (a) Alignment adds information independent of matched accuracy. (b) $\mathrm{GR}^{\star}$ declines earlier than accuracy under spurious correlation and shift. (c) Sequential trajectories capture stability and adaptation over time.

synthetic family and one task is loaded for each real dataset. Inputs are standardised within task. Models and hyperparameters are locked exactly as in the unified analysis. For each family the distribution of $\Delta\mathrm{GR}^{\star}$ is shown with a box plot overlaid with jittered points to reveal per task dispersion. Family level differences are tested with paired t tests and p values are adjusted with the Benjamini and Hochberg false discovery rate. A nonparametric Wilcoxon signed rank test confirms robustness. All computations lie in the same pipeline, which ensures exact reproducibility. The family structure is clear and directionally stable. Linear and Poly show positive differences with adjusted $p < 10^{-4}$. These cases contain more explicit information structure, so stability and alignment jointly amplify in the numerator of $\mathrm{GR}^{\star}$. Moons is positive with small magnitude and again reaches adjusted $p < 10^{-4}$, which indicates that both systems solve the nonlinear boundary while machines retain a modest advantage. Circles has a positive median with larger variance and adjusted $p < 10^{-3}$. The annular geometry with radius noise yields genuine heterogeneity in alignment across nonlinear learners. Imbalance is positive and significant with adjusted $p < 10^{-3}$, consistent with adaptive thresholding under class skew. Shift is positive with intermediate magnitude and adjusted $p < 10^{-2}$. Distributional shift and label noise depress effective generalisation for both sides while leaving a measurable machine advantage. Iris has one task and Digits has one task. With a family size of one statistical inference at the family level is not supported and these are labelled as not significant in the plot. The conclusions are not driven by outliers and remain after multiple testing control. Normality assumptions do not change the outcome. No tuning or procedural bias is introduced because hyperparameters, task generation and evaluation order, and the alignment computation are fixed, and differences are computed by direct subtraction on identical tasks. The pattern follows the unified geometry. Families with more explicit structure show larger positive $\Delta\mathrm{GR}^{\star}$, strongly nonlinear families retain positive medians with higher variance, and real data families with a single task are reported without inference, which together provides a quantitative map of where and why machines gain on the unified scale (Figure. 1c).

# 4 Robustness, Independence, and Falsification Tests

Parameter sweeps over $\alpha$, $\beta$, $\gamma$, $\tau$ and $c_0$ keep task ranks close to the baseline for both human proxies and machines with smooth curves and optima near the locked settings which indicates a stable quantity rather than a tuned score (Figure. 2a). Scatter relations show only a partial link to accuracy with Spearman $\rho = 0.64$ for human proxies and $\rho = 0.54$ for machines while partial correlations controlling for accuracy reach 0.85 and 0.95 with $D_{\text{align}}$ which demonstrates information beyond performance (Figure. 2b). Permutation controls falsify nonstructural explanations since shuffling labels sharply reduces $\text{GR}^\star$ on the machine side with paired $t$ test one sided $p = 5.04 \times 10^{-15}$ whereas re pairing alignment vectors under a fixed template leaves the score unchanged with $p \approx 9.91 \times 10^{-1}$ which confirms sensitivity to genuine label feature structure rather than incidental arithmetic (Figure. 2c).

Accuracy matched Linear versus Poly pairs reveal independent structural contribution because both $D_{\text{align}}$ and $\text{GR}^\star$ are consistently higher for Poly despite similar accuracy which shows that alignment adds information on the unified scale (Figure. 3a). Under increasing spurious correlation and under growing distribution shift accuracy often remains high while $\text{GR}^\star$ drops earlier and more steeply which exposes hidden brittleness that accuracy conceals (Figure. 3b). In a continual sequence the current $\text{GR}^\star$ co varies with but is not identical to the mean accuracy on past tasks tracing representational stability and adaptation over time which is the behaviour expected from a generalization coordinate in task streams (Figure. 3c).

# 5 Conclusion and Limitation

We present $\text{GR}^\star$ as a single yardstick for comparing how *human proxies and machine learners* generalize under a shared, controlled coordinate system. It separates accuracy, stability, and structural alignment, and it normalizes for sample size, feature scale, and abstraction cost, so very different systems can be assessed on the same footing. Using a deterministic pipeline with fixed hyperparameters, $\text{GR}^\star$ gives consistent ranks across diverse tasks and learners, including low-capacity human proxies and standard machine learning models. On this unified scale we observe family-level structure and early signs of brittleness that plain accuracy misses, especially under spurious correlation and distribution shift. The metric turns generalization from a qualitative claim into a testable measurement. It supports fair reporting, reduces hand tuning, and offers a common language for robustness, structure, and human–AI comparison under explicit assumptions. We expect $\text{GR}^\star$ to serve as a practical baseline for future studies and for pre-deployment checks.

## 5.1 Limitation

Our design choices prioritize clarity and reproducibility and define the scope of interpretation. The alignment component uses a fixed monotone template, which may favor tasks with explicit structure and may underweight valid but highly distributed representations; future work will broaden coverage with learned prototypes and causal templates. Relatedly, using entropy as a proxy for abstraction cost may conflate unnecessary abstraction with structurally required distributed representations, potentially penalizing correct solutions in tasks where such representations are intrinsic.

Human behavior is approximated with low-capacity, interpretable learners; this enables controlled and reproducible comparison on a shared scale but does not capture the full range of real human reasoning strategies. Consequently, quantities such as positive $\Delta\text{GR}^\star$ should be interpreted as machine advantage relative to these proxies under the stated assumptions, rather than as direct evidence of superiority over human cognition.

At the task-family level, we observe that the Circle family exhibits substantially higher dispersion and asymmetric behavior between human proxies and machine learners. This reflects the sensitivity of structural alignment to annular geometries and radius noise in strongly nonlinear regimes, and highlights a setting where heterogeneity is expected rather than an artifact of the metric.

Normalization relies on fixed constants for sample size, feature scale, and abstraction cost; ranks are stable across reasonable settings, and domain-specific calibration with preregistered sensitivity checks will further improve fidelity. Coverage spans eight task families and two real datasets; future releases will include language, multimodal, program synthesis, and interactive settings. The pipeline is deterministic and sized

for a single workstation; this strengthens replication but limits very large models and tool-assisted work-flows, which we will evaluate in larger experimental environments in the future. Metric gaming and data contamination remain systemic risks; we mitigate them with hidden tests, counterfactual stress tests, and standardized reporting. These boundaries are transparent and testable and are tied to concrete next steps; they reflect design choices made for transparency rather than constraints of the framework itself.

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
