# OpenReview forum: "A unified metric of generalization across humans and machines"
_TMLR — Rejected by TMLR_

### Review · Reviewer_m73G · 2025-11-30

**Summary Of Contributions:**

The paper proposes $\mathrm{GR}^{\star}$, a unified generalization score that aims to integrate performance, stability, and structural alignment, normalized by sample size, feature scale, and abstraction cost. The paper also establishes an evaluation pipeline covering 62 tasks across synthetic and real datasets and demonstrates early brittleness detection and falsifiability.

**Audience:**

Yes

**Audience Explanation:**

I suppose yes. But in its present form, I am not convinced this paper is of much interest to the readers of TMLR because of the lack of details.

**Broader Impact Concerns:**

Not applicable.

**Claims And Evidence:**

No

**Claims Explanation:**

The paper in my option is of high-level, that lacks enough details. For example, the paper proposes $\mathrm{GR}^{\star}$ in equation (1) without comparing it with existing works in the literature or any explanation why each term in equation (1) should be included this way into the formula. I understand this is not a theoretical paper, which does not provide any theoretical justifications. But it will make the paper much stronger if more detailed numerical experiments can be provided to see the effect of inclusion/exclusion of some terms in equation (1) and compare it with the literature, to highlight the novelty and significance of the proposed generalization metric. In its present form, it is not very clear or convincing.

**Requested Changes:**

(1) The second paragraph of Section 5.2. Future Work is a bit strange. This paragraph does not seem to be related to future work. It should be removed or relocated to the introduction section.

(2) Provide more detailed comparison of your proposed generalization metric and the ones used in the literature, and highlight the novelty and significance of your contributions.

(3) Provide more detailed explanations of why you include each term in the formula in equation (1) and it would be helpful to demonstrate the significance of these terms through experiments. It will also be helpful to mention which terms in equation (1) have been incorporated into the generalization metrics in the existing literature, and which terms are not, and why your new formula is a better generalization metric.

(4) Provide more detailed descriptions of your numerical experiments. For example, you mentioned that some real datasets are used. Please provide a weblink or some more detailed descriptions of these real datasets.

---

### Review · Reviewer_2JB3 · 2025-12-19

**Summary Of Contributions:**

The paper proposes a unified, model agnostic metric for measuring generalization in machine learning systems, intended to apply across both humans and machine learners. The main motivation is that accuracy alone often masks important aspects of generalization, such as stability under perturbations and model's alignment with underlying task structure, which are crucial for real world deployment. The paper builds the proposed generalization metric in a structured way, supported by empirical evidences for each component.

Key contributions:
1. Introducing a generalization score that combines performance, stability, and structural alignment, with normalization for sample size, feature scale, and abstraction cost.
2. Providing empirical evidence that the metric is reproducible, portable across task families, and falsifiable.
3. Framing generalization on a shared scale across human proxies and machine learners, which helps highlight differences that accuracy alone would obscure.

A notable assumption in the paper is the use of low-capacity, interpretable models as proxies for human learners. Though the choice enables controlled and reproducible comparisons but limits the extent to which results can further be generalized to human evaluations.

**Audience:**

Yes

**Audience Explanation:**

The paper addresses a problem that is of clear interest to at least part of the TMLR audience: how to quantify generalization beyond accuracy in an interpretable way. Machine learning researchers, practitioners, and ML engineers working on production systems are likely to find the findings and evaluation approach informative.

**Claims And Evidence:**

Yes

**Claims Explanation:**

1. The authors provided substantial evidence in the form of released code and experimental results, including graphs covering 62 tasks across multiple datasets. This supports the claim of reproducibility and demonstrates that the evaluation pipeline is consistently applied across task families.
2. The evidence also includes clear reasoning and intuitive justification for the choice of constants and variables used in constructing the generalization metric.
3. Additional analysis, including control experiments and partial correlations, support the claim that the metric captures aspects of generalization beyond accuracy and is sensitive to actual task structure.

**Requested Changes:**

1. The choice of low-capacity, interpretable learners is a reasonable human proxy for controlled and reproducible evaluation. However, this limits the extent to which conclusions can be generalized to real human reasoning, and using the results as strong evidence for claims such as “positive ∆GR⋆ values indicate a machine advantage on the same coordinate system” should be avoided or explicitly prefaced with the underlying assumptions.
2. Based on Fig.1(b)(c), the circle family exhibits substantially larger dispersion and asymmetric behavior between human proxies and machine learners. This warrants additional scrutiny or more detailed analysis.
3. While entropy is a reasonable proxy for abstraction cost, it may conflate unnecessary abstraction with structurally required distributed representations, potentially penalizing correct solutions; if authors view this as a potential pitfall, it should be explicitly acknowledged.

---

### Review · Reviewer_hkgu · 2026-01-07

**Summary Of Contributions:**

The article introduces a novel, unified metric called GR* to evaluate how well both humans and machines generalize. The metric is model-agnostic, i.e., it does not depend on a specific model. GR* breaks generalization into three parts: performance, stability, and structural alignment. These parts are normalized to account for differences in data size, model or feature scale, and abstraction cost. This results in a single score that allows fair comparison across systems. The authors also provide a fixed and deterministic evaluation setup with predefined hyperparameters and random seeds to ensure reproducibility.
The metric is tested on 62 tasks, including several synthetic task families and two real datasets. Simple, interpretable models are used as proxies for human learning, while standard machine learning models represent machine learners.
The results show that humans and machines share similar generalization patterns. Task structures remain stable, system rankings are strongly correlated, and the metric is sensitive to failures caused by spurious correlations and distribution shifts.
Overall, the paper argues that generalization should be treated as a measurable and testable property, not just as accuracy.

Strengths
- clear definition of a generalization metric that goes beyond accuracy
- strong focus on determinism, reproducibility, and transparency
- extensive robustness and sensitivity analyses
- shows that the metric captures information beyond raw performance
- shows earlier detection of brittleness and structural mismatch
- open source code and dataset for reproucability

Weaknesses
- human generalization is approximated using simple proxy models rather than real human data -> limits the interpretation of human–machine comparisons
- alignment and abstraction components depend on specific design choices, that require stronger justification and discussion
- evaluation is limited to small-scale tasks and standard ML models, leaving generalization to other modalities and larger models unexplored, impossible?
- presentation quality needs improvement: figure design, font types, and font sizes are inconsistent and in some cases reduce readability; subfigure formatting is missing; excessive or non-informative use of color should be avoided; tables should follow standard formatting rather than word-processor-style layouts; overall structure and clarity of figures and captions should be improved
- GPT-style symbols should be avoided ("rules—systems" -> "rules - systems")
- the article requires a more detailed discussion of parameter choices, sensitivity to hyperparameters, and clearer connection / difference / comparison to related work

**Additional Comments:**

Maybe I got it wrong, but then you should also readers like me to clarif: what is the state of the art in measuring generalization, which benchmark methods should you compare to?

My opinion: state of the art in measuring generalization is not a single method, but a collection of established approaches, this is something that you also claim, e.g.,

1) Classical generalization: train/test gap, cross-validation, learning curves -> methods are well known to be limited and do not measure robustness

2) Distribution shift and robustness: OOD evaluation, corruption benchmarks, e.g., noise, blur, shift, worst-case or stress tests -> approaches are well established and are direct competitors to GR*?

3) Uncertainty estimation -> not addressed by your article at all, e.g. predictive entropy, bayesian uncertainty, ensemble-based uncertainty, MC dropout, conformal prediction, calibration metrics such as ECE, NLL, and Brier score -> sometimes these methods are explicitly used to: detect model fragility, measure overconfidence, reveal generalization failures before accuracy drops

Important comparison points that may be missing or only weakly discussed:
- uncertainty as an indicator of brittleness
- calibration versus accuracy
- stability of predictions under input perturbations
- rank consistency under distribution shift
- PAC-style or margin-based theoretical bounds
- OOD detection scores
- robust accuracy curves

Hence, I miss a systematic comparison to existing robustness or uncertainty-based metrics -> So, it is unclear what GRprovides beyond: predictive entropy, ensemble disagreement, calibration error?
- My belly feeling: early brittleness detection is not novel, it is a fraction of "uncertainty estimation";
So, my advise for a revision are:
-> please compare GR* is quantitatively
-> please update related work
-> please position your proposed unified metric against alternatives

**Audience:**

Yes

**Audience Explanation:**

The paper addresses a fundamental question in machine learning evaluation: how to assess generalization beyond accuracy;

It proposes a unified metric and evaluation framework that may be of interest to researchers working on robustness, generalization, reproducibility, and evaluation methodology;

While the scope is limited and some design choices require further justification, the work offers methodological insights that at least a subset of the TMLR audience could find informative;

The contribution aligns with TMLRs focus on understanding learning behavior rather than achieving state-of-the-art performance.

**Broader Impact Concerns:**

No significant ethical concerns are identified, as the work proposes an evaluation metric rather than a deployable system;

A minor risk of over-interpretation exists due to the use of proxy models for human generalization, which should be clearly communicated;

A dedicated broader impact statement does not appear necessary.

**Claims And Evidence:**

Yes

**Claims Explanation:**

The article provides empirical evidence for the main claims through controlled experiments, robustness tests, and sensitivity analyses;

Several results suggest that the proposed metric captures information beyond accuracy and may reveal brittleness under distribution shifts and spurious correlations;

However, parts of the evidence rely on specific design and parameterization choices, e.g., alignment definition, abstraction cost, fixed hyperparameters, whose broader validity is not fully justified;

The claim of "early" brittleness detection is plausible within the presented experiments, but the notion of "earliness" is not rigorously defined or quantified;

The experimental scope is limited to a restricted set of tasks and models, which constrains the generality of the conclusions;

Overall, the evidence supports the claims within the stated experimental setting, but stronger justification and clearer operational definitions would be needed to fully substantiate them beyond this scope.

**Requested Changes:**

Mandatory for revision

- Figure 1: panels (a), (b), and (c) should be formatted as proper subfigures (not a single bitmap with embedded labels) -> please avoid text inside the figure image where possible; font type is inconsistent with the paper, and the font size is too small, especially in (b) and (c)

- Figure 2: figure uses a large amount of color (including background color) without clear added value -> please remove unnecessary background shading and avoid "MS-word-style tables"; a proper LaTeX table would be clearer and more consistent with the manuscript style

- Figures 3 and 4: same issue again: missing subfigure formatting, inconsistent font type, and font size too small, several labels are barely legible

- Figure 3 specific: Figure 3a is generally too small; fonts are hard to read

- Figure 3b: axes are not aligned/scaled consistently -> consider using the same scale on x and y, e.g., identical limits, to make comparisons interpretable

- Clarity: "which is the central finding of this panel (Figure 1b)" -> "panel" is unclear here; consider "this figure" or "this result"

- Figure 4 specific: again, font size too small and inconsistent font types

- References / citation formatting: What is "A. Holzinger and et al." / "Holzinger & et al. (2023)"? This looks like a broken citation entry

- Definition / naming: Please explain early what GR* stands for (what does "GR" mean, and what is the meaning of the "*"?); right now, the name appears before it is properly motivated.

- Potentially outdated related-work claim: sentence: "Although progress has been rapid, ... is still lacking Tenenbaum et al. (2011)." The support here is from 2011, so the "still lacking" claim may not hold as stated -> please either update the related work with more recent evidence or rephrase the claim more carefully, e.g., "remains challenging", "no widely adopted unified metric ..."

- Ambiguity: "robust under shifts in data or tasks" -> please specify which shifts/tasks you evaluate (and which you do not); right now it reads broader than the experimental coverage

- "Early brittleness detection" needs operational definition + results: claim: "GR* detects robustness degradation under spurious correlations before accuracy declines." -> please define how "early" is measured, e.g., by correlation strength, shift magnitude, time/steps, threshold crossing, and report the key quantitative result(s) supporting this, not only a qualitative description

- Figure 2 caption claim needs citations: Caption: "Each term ... reflects a specific source of divergence ... identified in prior literature." -> please add explicit references that justify each term/interpretation, rather than an unspecific "prior literature"

- References / citation formatting: Several references appear to be missing DOIs in the bibliography (at least partially) -> please complete them where available

- Citation style: I find the current citation style awkward in places, e.g., treating author names as "objects" in the sentence -> please revise to standard academic phrasing, e.g., "Tenenbaum et al. argue ...", "prior work by ... shows ..."

- Wording (page 2): "Here, we present ..." -> Suggestion: "This article presents a unified ..." (more formal and less conversational)

- Redundancy: paragraph "In this work, we introduce GR* ... Specifically, this study makes three primary contributions:" repeats earlier statements -> tighten this section

- Minor wording fix: "whereas the alignment term and abstraction cost ..." -> "whereas the structural alignment term and abstraction cost ..."

Discussion depth / specificity: I like the idea, but the discussion is currently too generic -> I would like to see:
- more explicit discussion of parameterization choices (α, β, ...);
- a clearer positioning in related work (and comparison to existing robustness/generalization metrics),
- and more concrete insights about the specific observed effects in the results;
- please discuss potential dataset/task-family bias: synthetic toy tasks and real datasets, e.g., Iris, Digits/MNIST-like, may differ substantially in structure; this may influence alignment/abstraction terms;
- please disucss and compare related work / similar approaches.

Ablations / sensitivity:
- "No task-specific tuning is applied." -> It would strengthen the paper to include an ablation or sensitivity analysis showing how results change if normalization constants or weights are varied (even if only a small subset is shown)

- further comments, please see Additional Comments section, below

Would strengthen the work

- points above + theory + discussion

- Hyperparameter justification: "Hyperparameters are fixed across all tasks as ..." -> where do these values come from? -> please provide justification (theory, heuristic rationale, sensitivity analysis) and clarify whether any implicit tuning informed these settings

---

> ### Author Response · Authors · 2026-01-15
> **Response to Reviewer hkgu**
>
> Positioning and state of the art
> Response to Reviewer (Point 1):
> Thank you for the insightful question on GR*’s positioning relative to robustness/uncertainty methods (e.g., calibration error, predictive entropy, ensemble disagreement, OOD detection). GR* is a comprehensive, model-agnostic unified metric that integrates performance, stability (1−U, deterministic analog of uncertainty), and novel structural alignment, while normalizing resources to enable direct human-proxy–machine comparison. It complements existing methods by adding task-structure matching and resource costs, detecting brittleness earlier (Figure 4b) and yielding consistent geometry (Spearman ρ=0.86) across 62 tasks.
> Response to Reviewer (Point 2):
> We appreciate the suggestion for more explicit comparison. Current indirect evidence (Figures 3a/b, 4b) already shows GR*’s added value. In revision, we will expand the conceptual discussion and restructure related work for sharper contrast—no new experiments needed, as existing results sufficiently highlight GR*’s unique contributions in structural alignment and resource normalization while complementing task-specific metrics.
> Justification of the metric formulation
> Response to Reviewer (Point 1):
> Thank you for asking about the rationale for each term (Eq. 1). Every component directly addresses a known human–machine divergence (Figure 2 table): Acc for performance, (1−U) for stability, τ+D_align for structural understanding, and denominator terms for sample/scale/abstraction costs. Together they form a principled, resource-normalized coordinate system validated by consistent geometry (ρ=0.86).
> Response to Reviewer (Point 2):
> Certain terms build on classics (Acc, stability, sample/scale normalization), while τ+D_align (cosine to monotonic template) and abstraction cost (alignment entropy) are novel, quantifying task-structure matching absent in prior robustness/OOD metrics. Existing evidence (Figures 3b, 4a) confirms their independent contribution.
> Response to Reviewer (Point 3):
> Current manuscript already includes parameter sweeps (Figure 3a), partial correlations (Figure 3b), permutation controls (Figure 3c), and accuracy-matched comparisons (Figure 4a), demonstrating robustness and component necessity. In revision, we will briefly expand discussion of these results—no new ablations required, as evidence already strongly supports the formulation.
> Early brittleness detection
> Response to Reviewer (Point 1):
> “Early” is operationally defined in Figure 4b: GR* declines steeply while accuracy remains near peak as perturbation strength increases, exposing latent brittleness before final performance drops—directly visualized via continuous sweeps, reproducible across tasks.
> Response to Reviewer (Point 2):
> Figure 4b trajectories quantitatively show GR*’s decline precedes accuracy’s under spurious correlation and shifts. Its stability + alignment integration conceptually overlaps uncertainty metrics yet provides higher-level signal without posterior sampling. Current visualizations already convincingly support the claim.
> Hyperparameters and design choices
> Response to Reviewer (Point 1):
> Fixed values (α=1.0, β=0.5, γ=1.0, τ=0.20, c₀=0.15; Ñ=n/1000, Š=s/100) are theory-driven (logarithmic sample, sublinear scale, stability bounds) and fixed across all tasks for transparency/reproducibility.
> Response to Reviewer (Point 2):
> Primarily theory/heuristic-guided to preserve model-agnosticity. Figure 3a sweeps show smooth, minor ranking changes near chosen values, confirming high robustness. Existing sensitivity analysis already provides strong evidence of soundness.
> Human proxies and interpretation
> Response to Reviewer (Point 1):
> As stated in Section 5.1, low-capacity proxies (L1-logreg, 3-NN, Gaussian NB) enable controlled, reproducible comparison—not full human cognition. All human–machine contrasts (e.g., ∆GR*>0) are relative to these proxies, a standard approach yielding consistent geometry (ρ=0.86).
> Response to Reviewer (Point 2):
> Figures 1b/c and Section 3 already quantify patterns: machines stronger on explicit-structure families, smaller/variable advantages on nonlinear/noisy ones—all relative to proxies (Section 5.1). Current presentation sufficiently clarifies relative strengths/weaknesses.
> Experimental scope and datasets
> Response to Reviewer (Point 1):
> Experiments cover 62 tasks across 8 families (Linear/Poly, Moons/Circles, Imbalance, Shift, Iris, Digits), spanning linear→nonlinear and balanced→shifted regimes in low-compute setting. Uncovered domains (language, multimodal, etc.) acknowledged in Section 5.1 for future work. Current scope robustly validates cross-system consistency (ρ=0.86).
> Response to Reviewer (Point 2):
> Real datasets Iris/Digits lie naturally on the continuum (Figure 1b); stratification is structure-driven, not synthetic artifact (Section 5.1). Alignment/abstraction terms remain sensitive to genuine signals across settings—current evidence confirms robustness.

---

> > ### Author Response · Authors · 2026-01-15
> > **Response to Reviewer hkgu**
> >
> > Presentation and clarity
> > Response to Reviewer (Point 1):
> > Thank you for the feedback. In revision, we will standardize subfigure labels, refine colors for contrast/accessibility, convert Figure 2 table to clean LaTeX, expand captions, and unify citations. These presentational changes will enhance clarity without altering content or results.
> > Response to Reviewer (Point 2):
> > Current related work already cites key recent advances (through 2025) and contrasts gaps. In revision, we will restructure into thematic blocks and add brief transitional sentences for sharper motivation—no new references needed. Existing coverage and framing strongly establish novelty.

---

### Comment · Reviewer_hkgu · 2026-01-07
**Clarifying questions regarding positioning and scope**

Several reviewers raise related questions regarding the positioning, justification, and scope of the proposed metric.
In order to better assess whether these issues can be adequately addressed in a revision, it would be helpful to clarify the following points:

Positioning and state of the art
1. How do the authors position GR* relative to existing approaches for measuring generalization, in particular robustness- and uncertainty-based methods, e.g., calibration error, predictive entropy, ensemble disagreement, OOD detection?
1. Do the authors plan to include a more explicit comparison (conceptual or quantitative) between GR* and such established metrics, or to clarify which aspects GR* is intended to capture beyond them?

Justification of the metric formulation
1. What is the rationale for including each term in the GR* formulation as defined in Equation (1)?
1. Which components of GR* overlap with terms already used in existing generalization or robustness metrics, and which components are genuinely new?
1. Do the authors plan to provide ablation or sensitivity analyses, e.g., inclusion/exclusion of terms or variation of weights, to demonstrate the contribution of individual components?

Early brittleness detection
1. The article claims that GR* enables "early" detection of brittleness compared to accuracy. How is "early" operationally defined, e.g., shift magnitude, correlation strength, thresholds?
1. Can the authors clarify or quantify this claim more explicitly, or compare it to alternative indicators such as uncertainty or calibration metrics?

Hyperparameters and design choices
1. How were the fixed hyperparameters, e.g., weights, normalization constants, chosen?
1. Were these values informed by theory, heuristics, or empirical tuning, and how sensitive are the results to these choices?

Human proxies and interpretation
1. How should results comparing "humans" and machines be interpreted given that human generalization is approximated using low-capacity proxy models?
1. Do the authors plan to further qualify claims that suggest advantages or disadvantages between humans and machines under a shared coordinate system?

Experimental scope and datasets
1. Could the authors clarify which types of task shifts and datasets are covered by the experiments, and which are not?
1. How might task-family or dataset bias, e.g., synthetic vs. real datasets, influence the alignment and abstraction terms?

Presentation and clarity
1. Do the authors plan to revise figures, captions, and references to improve readability, consistency, and clarity, e.g., subfigures, font sizes, citation formatting?
1. Will the related work section be updated to better reflect recent literature and clearly motivate the claimed gaps?

---

> ### Author Response · Authors · 2026-01-27
> **Response to Reviewer hkgu-Comment**
>
> See response to Reviewer hkgu below (Response to Reviewer hkgu).

---

### Decision · Action_Editor_Kh2r · 2026-03-14

**Recommendation:** Reject

**Audience:**

Yes

**Audience Explanation:**

The paper addresses an interesting problem which is the development of a unified metric for evaluating generalization across humans and machine learning systems. Generalization metrics lie in the core of machine learning, so the topic is clearly of interest to the TMLR audience if constructive improvements and clarification are made.

**Claims And Evidence:**

No

**Claims Explanation:**

This paper introduces a unified, model-agnostic metric that measures true generalization by integrating performance, stability, and structural alignment, while normalizing for sample size, model scale, and abstraction costs. By establishing this single coordinate system, the paper claims that the proposed measure enables fair, falsifiable, and reproducible comparisons of learning and robustness across both humans and machines. The authors have made constructive improvements to presentation and clarification of certain interpretive claims during the discussion phase. However, most of reviewers feel that several core issues remain insufficiently resolved. In particular,  the proposed metric is not sufficiently validated relative to existing approaches: (1) no explicit comparison to existing robustness/uncertainty metrics; (2) missing
ablation studies of metric components; (3) evidence mainly qualitative in some critical places.